# Monitoring of Land Desertification Changes in Urat Front Banner from 2010 to 2020 Based on Remote Sensing Data

**Yuanyuan Feng** [1,2,3], **Shihang Wang** [1,2,3,*], **Mingsong Zhao** [1,2,3] **and Lingmei Zhou** [1,2,3]

1   School of Geomatics, Anhui University of Science and Technology, Huainan 232001, China;
    austyyfeng@163.com (Y.F.); zhaomingsonggis@163.com (M.Z.); AUST_RSer@163.com (L.Z.)
2   Key Laboratory of Aviation-Aerospace-Ground Cooperative Monitoring and Early Warning of Coal
    Mining-Induced Disasters of Anhui Higher Education Institutes, Anhui University of Science and Technology,
    Huainan 232001, China
3   Coal Industry Engineering Research Center of Mining Area Environmental and Disaster Cooperative
    Monitoring, Anhui University of Science and Technology, Huainan 232001, China
*   Correspondence: 2011058@aust.edu.cn

**Abstract:** Monitoring the spatio-temporal dynamics of desertification is critical for desertification control. Using the Urat front flag as the study area, Landsat remote sensing images between 2010 and 2020 were selected as data sources, along with MOD17A3H as auxiliary data. Additionally, RS and GIS theories and methods were used to establish an Albedo–NDVI feature space based on the normalized difference vegetation index (NDVI) and land surface albedo. The desertification difference index (DDI) was developed to investigate the dynamic change and factors contributing to desertification in the Urat front banner. The results show that: ① the Albedo–NDVI feature space method is effective and precise at extracting and classifying desertification information, which is beneficial for quantitative analysis and monitoring of desertification; ② from 2010 to 2020, the spatial distribution of desertification degree in the Urat front banner gradually decreased from south to north; ③ throughout the study period, the area of moderate desertification land increased the most, at an annual rate of 8.2%, while the area of extremely serious desertification land decreased significantly, at an annual rate of 9.2%, indicating that desertification degree improved during the study period; ④ the transformation of desertification types in Urat former banner is mainly from very severe to moderate, from severe to undeserted, and from mild to undeserted, with respective areas of 22.5045 km$^2$, 44.0478 km$^2$, and 319.2160 km$^2$. Over a 10-year period, the desertification restoration areas in the study area ranged from extremely serious desertification to moderate desertification, from serious desertification to non-desertification, and from weak desertification to non-desertification, while the desertification aggravation areas ranged mainly from serious desertification to moderate desertification; ⑤ NPP dynamic changes in vegetation demonstrated a zonal increase in distribution from west to east, and significant progress was made in desertification control. The change in desertification has accelerated significantly over the last decade. Climate change and irresponsible human activities have exacerbated desertification in the eastern part of the study area.

**Keywords:** urat front flag; desertification; spatio-temporal pattern; driving factor; albedo–ndvi; npp

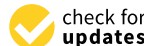



## 1. Introduction

Desertification is a global ecological problem [1]. China has the world's largest desertification area, the most affected population, and the most severe wind and sand hazards, with the desertification land area reaching 1533 million km$^2$, accounting for 15.9% of the total national land area [2]. Inner Mongolia Autonomous Region is located on China's northern border, and is the largest and most comprehensive ecological function area in the north of China. It is also one of the provinces with the highest concentrations of desertification and sandy land, as well as the most severe harm, with the majority of the region being in arid, semi-arid, and subtropical arid climate zones. Within the distribution area of

Badangilin, Tengger, Ulanbu, and Kubuchi's "four deserts" and Mawusu, Hunshandake, Horqin, and Hulunbeier's "four sands", desertification land covers 914 million mu, accounting for 23.3% of the national desertification area, the area of sandy land is 612 million mu, accounting for 23.7% of the national sandy land area, and the ecological environment is extremely fragile [3–5]. At the moment, as a significant ecological barrier along China's northern border, the impact of human activities on the ecological environment has exacerbated land desertification, severely impeding the healthy and sustainable development of human society and the improvement of the ecological environment suitable for human habitation [6,7]. As a result, scientifically sound and accurate data on the desertification status can serve as a critical foundation for desertification research and control [8].

Thus far, several researchers have used remote sensing and GIS techniques in their research on desertification [9–11]. F. Basso et al. assessed the environmental sensitivity of the Agri basin in southern Italy at the watershed scale using GIS and remote sensing data [12]. Jeong et al. examined the change process of the desert transition zone in the Asian region from 1982 to 2008 and concluded that Asia is more vulnerable to desertification disturbances under global or regional warming conditions [13]. Salvati et al. evaluated the extent of land desertification in Italy based on the ESAI framework [14]. Wang Fei et al. and Zhang Jianxiang et al. analyzed the causes of land desertification in the Tarim Basin and Loess Plateau region [15,16]. Xue et al. analyzed the dynamic change characteristics of wind-eroded desertification areas in northwestern Shanxi using a transfer matrix [17]; Chen Fang et al. used MODIS data to monitor the degree of desertification in Mongolia dynamically [18]. In addition, the widely used desertification evaluation indicators in current remote sensing monitoring of desertification are vegetation cover, net primary productivity, surface albedo, soil texture, crop yield, and water erosion [19–21]. After a thorough examination of the causes, development process, and manifestations of desertification formation, it was determined that vegetation is the most active and significant influencing factor on the surface, as well as the most sensitive to changes in environmental factors such as topography, landform, soil, hydrology, and climate, which can be used as the primary basis for remote sensing monitoring of desertification [22]. For example, Sternberg et al. used NDVI to evaluate the process of land desertification on the Mongolian plateau [23]. Ying et al. used Albedo–NDVI feature space to assess the desertification of the Loess Plateau [24]. Lei et al. estimated the vegetation cover using an image element dichotomous model on the MOD13Q1-NDVI dataset and then examined the status of land desertification in Kenya [25]. Muzetijiang-Abla et al. analyzed the degree of desertification and spatial and temporal changes in the Loess Plateau region using GIMMS AVHRR NDVI data, concluding that the majority of the Loess Plateau is covered by moderate to severe desertification area, with the degree of desertification generally decreasing [26]. Zhu et al. proposed that the different vegetation types have a different maximum light energy utilization and developed a regional NPP estimation model using the Inner Mongolia vegetation type as an example. The regional NPP estimation model estimated vegetation net productivity and analyzed its spatial and temporal distribution characteristics [27]. However, the majority of previous studies used MODIS–NDVI or GIMMS–NDVI models to analyze desertification by calculating vegetation cover, while a few studies used an Albedo–NDVI feature space approach for desertification in the Urat front flag. However, the research data are outdated and unrepresentative. The majority of research focuses on the influencing factors of desertification, desertification evaluation indexes, and the monitoring and comparison of desertification degrees in different periods. There are insufficient studies on the rapid positioning, identification, and spatial distribution of desert areas formed by the land degradation process under long time series, as well as the problems of strong subjectivity of research results, difficulty in realization, low precision, and complexity of methods. Desertification appears as a bare land surface in remote sensing images. The enhancement of information and the weakening of vegetation information can be characterized by index factors such as surface albedo, surface temperature, surface humidity, vegetation index, and vegetation cover. The normalized vegetation index (NDVI) is a crucial biophysical

parameter reflecting the state of surface vegetation, whereas the surface albedo is a physical parameter representing the reflection characteristics of the surface to solar short-wave radiation. As desertification increases, surface vegetation is severely damaged, surface vegetation cover decreases, biomass drops, and surface roughness increases, which is reflected in remote sensing images by a corresponding decrease in NDVI value and an increase in surface albedo. Therefore, it is highly representative to research and analyze the degree of desertification in the region using "Albedo–NDVI" feature space.

The dry climate, scarcity of precipitation, temperate continental arid climate, and excessive impact of human activities on the ecological environment have resulted in the ecological environment of the Urat front flag facing problems with soil salinization, sanding, and pasture degradation, weakening the ecological environment [28,29]. According to previous desertification research, combining the vegetation index and surface albedo by constructing an "Albedo–vegetation index (NDVI) feature space" enables more effective and convenient quantitative monitoring and research of the spatial and temporal distribution and dynamics of desertification [30]. In addition, it can serve as a theoretical basis for ecological restoration and environmental management in the region. In light of this, to analyze the spatial and temporal characteristics and evolution patterns of land desertification in the Urat front flag, this study created the region's "Albedo–NDVI" feature space between 2010 and 2020 using the ENVI and ArcGIS software platforms. Additionally, using Landsat remote sensing images as the primary data source, as well as dynamic attitude calculation [31], transfer matrix analysis [32], and center of gravity shift analysis [33], and in conjunction with MODIS–NDVI data, the CASA model was used to estimate the net primary productivity of vegetation between 2010 and 2020 [27,34]. To accurately and objectively evaluate the ecological environment of the study area, this paper analyzes the spatial distribution pattern and the impact of topographic factors on desertification. It also provides a theoretical basis for the ecological environment governance of the Urat front banner, and serves as a reference for the ecological environment protection and rational development of Hetao oasis and arid and semi-arid areas.

## 2. Materials and Methods

### 2.1. Overview of the Study Area

The Urat front banner is located in the eastern part of the Loop Plain, where the administrative division is under Bayannur City in the Inner Mongolia Autonomous Region. The geographical coordinates are $108°11'$–$109°54'$ E and $40°28'$–$41°16'$ N. The climate is temperate continental, with an annual average temperature of 6–7 °C, annual sunshine hours of 2959.4–3456.7 h, annual evaporation of 2069.3–2365.3 mm, and annual precipitation of 200–250 mm, mostly concentrated in June–September, accounting for 78.9% of the annual precipitation. The total area of the district is 7476 km$^2$, with the yellow irrigation area covering $7.47 \times 10^4$ hm$^2$, accounting for 54.55% of the arable land area. Specific location information is shown in Figure 1.

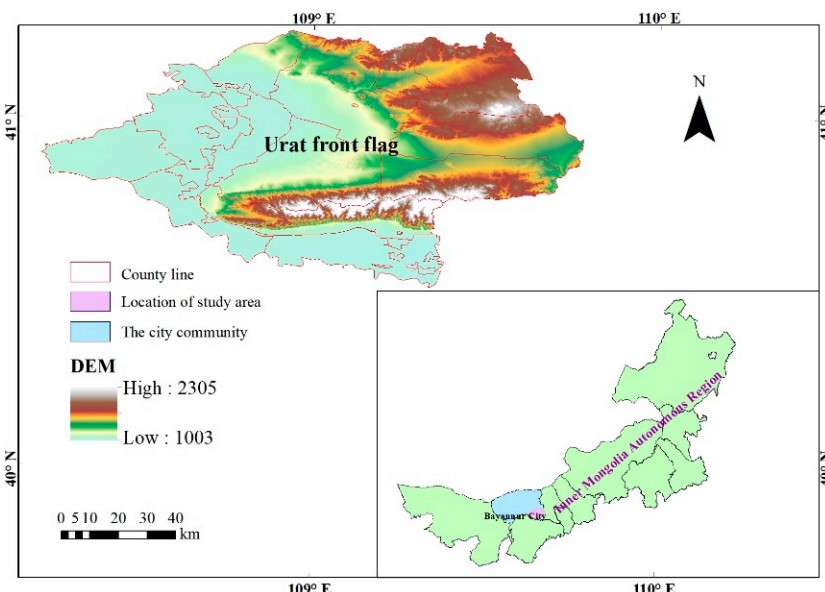

**Figure 1.** Geographical position map of the Urat front flag.

### 2.2. Data Source and Its Pre-Processing

Landsat images were obtained from the USGS [35], with time periods of 27 August 2010 and 7 September 2020, respectively. The selected time period has low cloud cover, and the image quality meets the research requirements. The remote sensing images were preprocessed using ENVI5.3 software, which took into account radiometric calibration, atmospheric correction, and geometric refinement correction, while keeping the error to less than one image element. Finally, the remote sensing images were stitched and cropped based on the vector boundaries of the Urat front banner, and relevant parameters were inverted on the preprocessed remote sensing images [36,37]. The normalized vegetation index (NDVI) is a biophysical parameter representing the surface vegetation status, whereas surface albedo reflects the physical characteristics of the surface. By combining these two indicators to extract desertification information together and avoiding the use of a single spectral information classification, quantitative dynamic monitoring of desertification can be made more convenient and effective.

The MODIS/Terra MOD17A3H data products [38] provide accurate measurements of terrestrial vegetation growth, including global gross primary productivity (GPP) and annual net primary productivity (NPP). The NPP data were synthesized from 45 periods (8-day synthetic data) at a spatial resolution of 500 m in $gC/m^2$, and were processed using projection change, mosaicking, and cropping to investigate the spatial and temporal variation of NPP in the two years of 2010 and 2020 in the Urat front banner, as well as to conduct the driving factor analysis for this study [39,40]. Negative values, fire point instability values, and background noise are not removed from the monthly products of NPP, but the annual data for 2010 and 2020 provide cloud-free mean lights and exclude transient lights, so the use of MOD17A3 data for these two years to validate only the NPP estimated by the CASA model is sufficient for the experiment.

### 2.3. Research Method and Process

In view of the principle of invertibility, reliability, and practicability of the desertification degree discriminant index [41], vegetation parameters were selected as the discriminatory index of desertification degree. Additionally, the normalized difference vegetation index (NDVI) and the land surface albedo (Albedo) were obtained, which characterize the biophysical characteristics of surface vegetation state and soil information, respectively. Furthermore, using the MOD17A3 data obtained for the two years of 2010 and 2020, vegetation annual NPP values were calculated using the CASA model to assess desertification genesis.

### 2.3.1. Normalized Difference Vegetation Index (NDVI)

The normalized vegetation index (NDVI) was calculated using reflectance values in the near-infrared and red wavelength bands [42]. Additionally, the NDVI value, *N*, was normalized using the following formula to facilitate subsequent data comparison and feature space construction.

$$\text{NDVI} = (\rho_{\text{nir}} - \rho_{\text{red}}) / (\rho_{\text{nir}} + \rho_{\text{red}}) \tag{1}$$

$$N = (\text{NDVI} - \text{NDVI}_{\text{min}}) / (\text{NDVI}_{\text{max}} - \text{NDVI}_{\text{max}}) \tag{2}$$

where $\rho_{\text{nir}}$ denotes the reflectance in the near-infrared band, $\rho_{\text{red}}$ indicates the reflectance in the red band, and $\text{NDVI}_{\text{max}}$ and $\text{NDVI}_{\text{min}}$ represent the maximum and minimum values of NDVI.

### 2.3.2. Land Surface Albedo

The surface albedo was inverted using the inversion model established by Liang [43], and the Albedo value, *A*, was normalized using the following equation.

$$\text{Albedo} = 0.356 \times \rho_{\text{blue}} + 0.130 \times \rho_{\text{red}} + 0.373 \times \rho_{\text{nir}} + 0.085 \times \rho_{\text{swir1}} + 0.072 \times \rho_{\text{swir2}} - 0.0018 \tag{3}$$

$$A = (\text{Albedo} - \text{Albedo}_{\text{min}}) / (\text{Albedo}_{\text{max}} - \text{Albedo}_{\text{max}}) \tag{4}$$

where $\rho_{\text{blue}}$ indicates the reflectance of the blue band, $\rho_{\text{red}}$ denotes the reflectance of the red band, $\rho_{\text{nir}}$ represents the reflectance of the near-infrared band, $\rho_{\text{swir1}}$ and $\rho_{\text{swir2}}$ show the reflectance of two mid-infrared bands, and $\text{Albedo}_{\text{max}}$ and $\text{Albedo}_{\text{min}}$ are regarded as the maximum and minimum values of Albedo, respectively.

### 2.3.3. Albedo–NDVI Feature Space Analysis

According to the findings of Y. Zeng et al. [30], there is a significant linear negative correlation between NDVI and albedo in the one-dimensional feature space, and the distribution of different surface cover types in the Albedo–NDVI feature space exhibits a significant divergence pattern, allowing for easy differentiation of various surface cover types [44,45] (Figure 2).

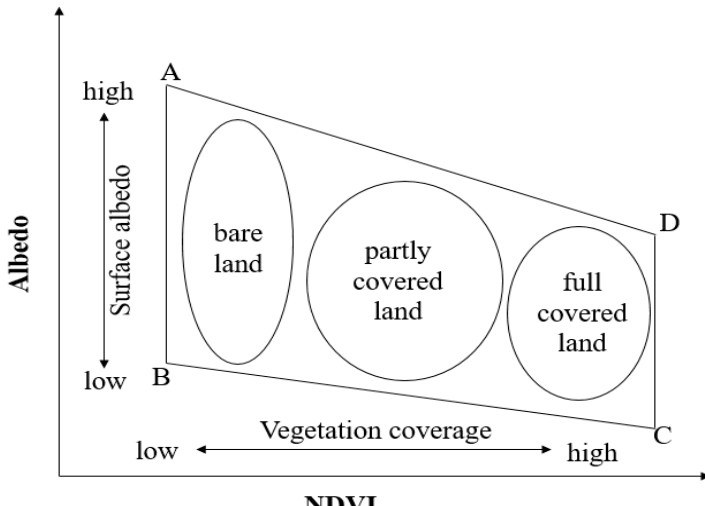

**Figure 2.** Albedo–NDVI feature space. Note: A is arid bare soil (low NDVI and high Albedo), B is water-rich bare soil (low NDVI and low Albedo), C is high vegetation cover with adequate soil moisture content (high NDVI and low Albedo), and D is high vegetation cover area with low soil moisture content and relatively high albedo (low NDVI and high Albedo).

Surface albedo is a function of vegetation cover and soil water content in the Albedo–NDVI space. The top boundary AD in Figure 2 represents the high albedo line, reflecting the drought condition, and the bottom edge BC is the maximum low albedo line, representing the condition of sufficient surface moisture [30].

The linear regression equation between NDVI and Albedo was constructed as follows based on their linear negative correlation:

$$\text{Albedo} = k \times \text{NDVI} + b \tag{5}$$

where $k$ is determined by the slope of the fitted curve in the feature space.

### 2.3.4. Desertification Difference Index (DDI)

The degree of desertification is not only a direct indicator of the severity of land desertification, but also an indirect indicator and measure of the ease with which desertification land can be restored to its productive and ecosystem functions [46]. The DDI [47] is expressed as follows in terms of Albedo–NDVI feature space:

$$\text{DDI} = a \times \text{NDVI} - \text{Albedo} \tag{6}$$

### 2.3.5. Accuracy Verification of Desertification Classification

This study employs confusion matrix analysis to determine the classification accuracy and reliability, which can indicate not only the total error for each category, but also the category misclassification. The error matrix is composed of the number of image elements classified into a category and a proportional array containing the high-resolution relative test truth value for that category [48,49]. The confusion matrix can be used to calculate the overall accuracy (OA), producer's accuracy (PA), user's accuracy (UA), and Kappa coefficient evaluation metrics, which are used to compare various aspects of desertification recognition accuracy. The PA is the probability of a sample point being correctly classified, which is a measure of missing error. Additionally, UA quantifies the inclusion error by comparing the ratio of the correctly classified sample points in each row to the total sample points in that row. Theoretically, its formula is expressed as follows:

$$\text{OA} = \sum_{i=1}^{i=5} X_{ii} / N \tag{7}$$

$$\text{PA} = X_{ii} / X_{i+} \tag{8}$$

$$\text{UA} = X_{ii} / X_{+i} \tag{9}$$

$$\text{Kappa} = \frac{\left[ N \sum_{i=1}^{i=5} X_{ii} - \left( \sum_{i=1}^{i=5} X_{+i} + X_{+i} \right) \right]}{N^2 - \sum_{i=1}^{i=5} X_{+i} + X_{+i}} \tag{10}$$

where $N$ indicates the total sample count, $X_{ii}$ denotes the sample count in row $i$ and column $i$, and $X_{i+}$, and $X_{+i}$ represent the total sample counts of row $i$ and column $i$, respectively.

To ensure the accuracy of the results, 100 validation points were distributed uniformly and randomly throughout the experimental area. The validation points were visually interpreted based on Landsat 8 true color images and Google Earth maps. The resulting interpretation data were compiled to construct a confusion matrix and solve for the Kappa coefficient.

### 2.3.6. Desertification Land Transfer Matrix Model

The transfer matrix model in ArcGIS 10.5 was used to analyze the temporal dynamics of different classes of desertification land. The transfer matrix was calculated as [50,51]:

$$D_{ij} = S_{ij} / \sum_{i=1}^{n} \sum_{j=1}^{n} S_{ij} \tag{11}$$

where $i$ and $j$ denote desertification land grades, $D_{ij}$ represents the transfer degree of different desertification land grades, $S_{ij}$ indicates the transfer area of different desertification land grades (km$^2$), and $n$ is regarded as the total amount of desertification land of the same grade.

### 2.3.7. Dynamic Change Model of Desertification Land

To further reflect the annual rate of change in the land desertification area, the dynamic attitude, $K$, is used to quantify the rate of change in desertification [52].

$$K = \frac{u_2 - u_1}{u_1} \times \frac{1}{t_2 - t_1} \times 100\% \tag{12}$$

where $K$ denotes the dynamic attitude of a desertification land type over the study period (%), $u_1$ represents the initial area (km$^2$), $u_2$ indicates the final area (km$^2$), and $t_1$ and $t_2$ indicate the time corresponding to the initial and final areas, respectively.

### 2.3.8. Center of Gravity Migration Model of Desertification Land

The migration of the spatial center of gravity can be used to characterize the pattern of spatial change in a landscape. The principle of population geography's center of gravity [53,54] is used to analyze the direction and distance of desertification land's center of gravity migration, as well as the spatial dynamic changes associated with various grades of desertification land. The coordinates of a given grade of desertification land's center of gravity in year $t$ are calculated using the following formula [55,56]:

$$X_t = \sum_{i=t}^{n} C_{ti} \times X_i / \sum_{i=t}^{n} C_{ti} \tag{13}$$

$$Y_t = \sum_{i=t}^{n} C_{ti} \times Y_i / \sum_{i=t}^{n} C_{ti} \tag{14}$$

where $X_t$ and $Y_t$ represent the latitude and longitude coordinates of the center of gravity, respectively, for the distribution of a certain type of desertification land in year $t$, $C_{ti}$ denotes the area (km$^2$) of the ith patch of a certain type of desertification in year $t$, and $X_i$ and $Y_i$ indicate the latitude and longitude coordinates of the center of gravity, respectively, for the distribution of the ith patch of a certain type of desertification in year $t$.

$$V_{(t2-t1)} = \frac{\sqrt{[x_{t2} - x_{t1}]^2 + [y_{t2} - y_{t1}]^2}}{t_2 - t_1} \tag{15}$$

where $V_{(t2-t1)}$ indicates the migration rate of the center of gravity (m/a) for a certain type of desertification land in a certain time period, $t_2$ and $t_1$ denote the study termination and initiation times, respectively, and $x_{t1}, y_{t1}, x_{t2}$, and $y_{t2}$ represent the latitude and longitude coordinates of the center of gravity for the distribution of a certain type of desertification land in year $t_1$ or $t_2$, respectively [57].

### 2.3.9. CASA Model

The light energy utilization model to estimate NPP is based on a resource balancing view and is one of the most well-documented inversion models for NPP remote sensing. The Carnegie–Ames–Stanford approach (CASA) model, established by Potter et al. [58], is the most widely used model for light energy utilization.

$$\text{NPP}(x,t) = \text{APAR}(x,t) \times \varepsilon(x,t) \tag{16}$$

where $x$ denotes spatial position, $t$ represents time, APAR($x,t$) indicates the photosynthetically active radiation (gC·m$^{-2}$·month$^{-1}$) absorbed by image element $x$ in month $t$, and $\varepsilon(x,t)$ is regarded as the actual light energy utilization (gC·MJ$^{-1}$) of image element $x$ in month $t$.

$$\text{APAR}(x,t) = \text{SOL}(x,t) \times \text{FPRA}(x,t) \times 0.5 \tag{17}$$

where SOL indicates the total solar radiation (gC·m$^{-2}$·month$^{-1}$), FPRA represents the proportion of the incident photosynthetically active radiation absorbed by the vegetation layer, and the constant 0.5 denotes the ratio of solar radiation utilized by the vegetation to the total solar radiation [34,59].

## 3. Results and Analysis

### 3.1. Albedo–NDVI Feature Space Analysis

The ROI function in ENVI was used to randomly select 1600 sample points distributed at various desertification levels across the study area. After normalizing the data of the two periods, the NDVI and Albedo values were extracted from the 1600 sample points, and a linear regression equation was constructed between them. The regression equations and linear correlations in the study area for 2010 and 2020 (Figure 3) were as follows:

$$\text{Albedo} = -0.1505\text{NDVI} + 0.3855 \left( R^2 = 0.7004 \right) \tag{18}$$

$$\text{Albedo} = -0.098\text{NDVI} + 0.2190 \left( R^2 = 0.7033 \right) \tag{19}$$

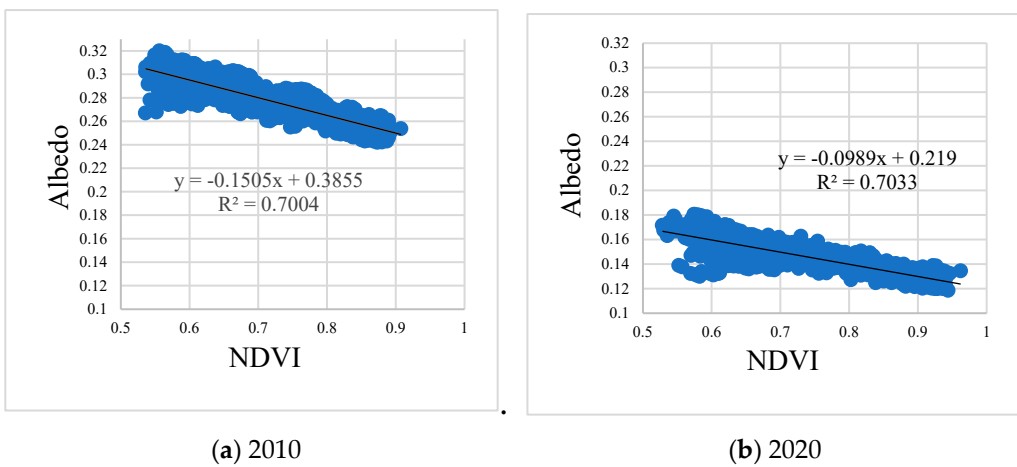

(**a**) 2010　　　　　　　　　　　　　　　　(**b**) 2020

**Figure 3.** Albedo-NDVI linear regression analysis.

As illustrated in Figure 3, the Albedo–NDVI feature space is trapezoidal, and the R$^2$ coefficients of linear regression equations are all greater than 0.7, and Albedo$_{max}$ is significantly negatively correlated with NDVI. Meanwhile, as the vegetation index gradually increases, its surface albedo decreases, demonstrating a strong linear negative correlation between the vegetation index and surface Albedo.

### 3.2. DDI Analysis

The DDI was estimated for all years using the Albedo–NDVI feature space (Figure 3). To this end, $k$ values were calculated using $a \times k = -1$, and the final DDI expressions were obtained for the two phases of the Urat front flag data as reported in Equations (17) and (18).

$$\text{DDI}_{2010} = 6.6445 \times \text{NDVI} - \text{Albedo} \tag{20}$$

$$\text{DDI}_{2020} = 10.2041 \times \text{NDVI} - \text{Albedo} \tag{21}$$

Desertification is characterized using DDI, which is classified into five categories based on the natural break method, namely extremely serious desertification, serious desertification, moderate desertification, weak desertification, and non-desertification [60,61].

### 3.3. Accuracy Validation

Classification error analysis using the Albedo–NDVI feature space (Table 1) revealed that the overall accuracy of both the 2010 and 2020 images was over 95%, with Kappa

coefficients of 0.96 and 0.93, meeting the requirements for desertification dynamics research in the region. Among them, the Albedo–NDVI model has high production accuracy for moderate, serious, and extremely serious desertification, implying that underestimation of these three desertification levels is minimal. However, the production and user accuracy for weak desertification is extremely low, indicating a high multimetric error in identifying weak desertification areas.

**Table 1.** Error analysis of land desertification information extraction based on Albedo-NDVI feature space.

| Year | Extremely Serious (%) | | Serious (%) | | Moderate (%) | | Weak (%) | | Non-Desertification (%) | | Kappa Coefficient | OA (%) |
|------|------|------|------|------|------|------|------|------|------|------|------|------|
| | PA | UA | PA | UA | PA | UA | PA | UA | PA | UA | | |
| 2010 | 97.14 | 97.29 | 91.59 | 97.67 | 98.08 | 92.67 | 98.32 | 85.18 | 97.63 | 99.69 | 0.96 | 96.27 |
| 2020 | 99.98 | 100.00 | 98.45 | 98.62 | 95.57 | 87.83 | 82.10 | 65.73 | 81.38 | 99.77 | 0.93 | 95.06 |

The validation results demonstrate that the Albedo–NDVI feature spatial method is capable of achieving optimal results in both the extraction of desertification land and the classification of desertification degree with higher accuracy and efficiency than other widely used remote sensing methods for desertification monitoring.

### 3.4. Spatial Distribution Characteristics of Desertification Land

The Urat front flag is a significant environmentally sensitive and ecologically fragile area in Inner Mongolia due to its dense population distribution and the high level of disturbance caused by human activities. To thoroughly investigate the spatial and temporal evolution of desertification in the Urat front flag, it is necessary to examine the area's desertification process.

Desertification information was retrieved using the aforementioned methodologies and indicators in ArcGIS 10.5 software, and finally, the desertification land grade map of the study area was generated for two time periods of 2010 and 2020, as shown in Figure 4.

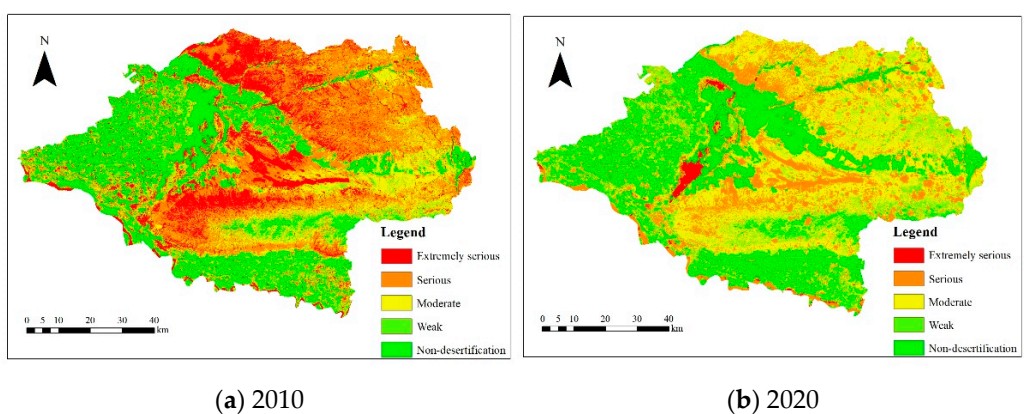

(**a**) 2010                                                      (**b**) 2020

**Figure 4.** Spatio-temporal distribution of desertification grade in the Urat front banner based on DDI.

As seen in Figure 4, the spatial distribution of desertification in the Urat front flag 10 years from 2010 to 2020 demonstrates a downward trend from south to north. In 2010, extremely serious desertification land was primarily distributed in the central and northern parts (e.g., Erdenbrag Sumu and Bayannur City Dashetai Ranch), serious desertification land was mainly distributed on the northeast (e.g., Xiaoshetai Town and Mingan Town), moderate desertification land gradually expanded from the south to the west, weak desertification land was mainly distributed on the western and southern areas (e.g., Bayannur City Xin'an Farm, Bayannur City Zhongtan Farm, and surrounding areas), and non-desertification land was mainly concentrated in the central and western areas (e.g.,

Bayannur City Uliang Suhai Fishing Ground, Sudulun Town, and Bayannur City Sudulun Farm). Comparing 2020 to 2010 reveals that the desertification degree is obviously reduced, primarily as a result of the implementation of the policy of "returning farmland to forest and grass and replacing it with food", which has increased the vegetation cover in the region and significantly enhanced the quality of the ecological environment. However, after 10 years of development, the moderate desertification and non-desertification lands in the Urat front flag are expanding, mainly to the southwest and northeast, while a part of the non-desertification land in the central area has shifted to very serious desertification. Additionally, the development of weak desertification land is mainly concentrated in the center and south, and the changes of extremely serious desertification and serious land have declined sharply.

### 3.5. Characteristics of Desertification Land Area Change

The changes in the total area of various types of desertification land reflect the general development trend of desertification land in the study area. The statistical results for the area of various types of desertification land in the study area in the two periods of 2010 and 2020 are reported in Table 2.

**Table 2.** Types and area of desertification land in the Urat front banner, $km^2$.

| The Type of Desertification Land | 2010 | | 2020 | | 2010–2020 |
|---|---|---|---|---|---|
| | Area | % | Area | % | Annual Rate of Change (%) |
| Extremely serious | 1236.1077 | 16.6 | 95.0265 | 1.3 | −9.2 |
| Serious | 2253.2328 | 30.2 | 1400.8446 | 18.8 | −3.8 |
| Moderate | 1535.6655 | 20.6 | 2790.1539 | 37.4 | 8.2 |
| Weak | 1042.2000 | 13.9 | 1471.8825 | 19.7 | 4.1 |
| Non-desertification | 1395.3690 | 18.7 | 1704.9951 | 22.8 | 2.2 |
| Sum | 7462.5750 | 100.0 | 7462.9026 | 100.0 | 1.5 |

The changes in the total area of various types of desertification land reflect the general development trend of desertification land in the study area. The statistical results for different types of desertification land areas in the study area in the two periods of 2010 and 2020 are reported in Table 2. As can be observed, the land area of the Urat front flag expanded from 7462.5750 $km^2$ to 7462.9026 $km^2$ between 2010 and 2020, increasing the total area by 0.3276 $km^2$. Among these, the areas of extremely serious and serious desertification land were both reduced to varying degrees, from 16.6% and 30.2% of the total desertification land, respectively. The area of moderate desertification land increased from 1535.6655 $km^2$ in 2010 to 2790.1539 $km^2$ in 2020, while the area of weak desertification land increased from 13.9% to 19.7%. Additionally, the area of non-desertification land increased from 1395.3690 $km^2$ to 1704.9951 $km^2$, with the proportion increasing from 18.7% to 22.8%. As can be seen, the desertification land in the study improved overall throughout the 10-year period from 2010 to 2020, with the degree of desertification reversing from serious and extremely serious to moderate and weak, resulting in a spike in moderate desertification.

Between 2010 and 2020, the dynamic attitudes of extremely serious and serious desertification areas were −9.2% and −3.8%, respectively, indicating that extremely serious and serious desertification areas decreased by 9.2% and 3.8% per year, respectively, while the moderate, weak, and non-desertification areas increased by 8.2%, 4.1%, and 2.2%, respectively. Throughout the study period, the area of moderate desertification land increased the most, whereas the area of extremely serious desertification land decreased the most, demonstrating that desertification improved throughout the study period.

### 3.6. Characteristics of Desertification Land Transfer Change

The transfer matrix efficiently expresses the interconversion relationship between different types of land across the two periods of 2010 and 2020, and can be used to investigate the spatial evolution process and characteristics of desertification land. Using the mathematical model of transfer matrix, the transfer matrix of desertification land from 2010 to 2020 was obtained as reported in Tables 3 and 4 and spatially depicted in Figure 5.

**Table 3.** Transition matrix of desertification in the Urat front banner from 2010 to 2020, km$^2$.

| 2010 \ 2020 | Extremely Serious | Serious | Moderate | Weak | Non-Desertification | Total (Reduced) |
|---|---|---|---|---|---|---|
| **Extremely serious** | 27.5886 | 695.0610 | 420.8040 | 46.5273 | 45.8181 | 1235.7990 |
| **Serious** | 16.1136 | 433.18000 | 1476.2150 | 220.0040 | 107.3880 | 2252.9010 |
| **Moderate** | 22.5045 | 176.7820 | 662.2933 | 470.4828 | 203.3020 | 1535.3650 |
| **Weak** | 18.4590 | 51.3909 | 141.0400 | 415.2340 | 415.8430 | 1041.9670 |
| **Non-desertification** | 10.3428 | 44.0478 | 89.3223 | 319.2160 | 932.2280 | 1395.1570 |
| **Total (increased)** | 95.0085 | 1400.4620 | 2789.6750 | 1471.4640 | 1704.4790 | 7461.1880 |

**Table 4.** The change in the spatiotemporal pattern of desertification in the study area, km$^2$.

| 2010–2020 | Severe Deterioration | Deterioration | No Change | Restoration | Obvious Restoration |
|---|---|---|---|---|---|
| Area | 2470.524 | 236.0673 | 653.1516 | 1625.8000 | 1043.8430 |
| Percent (%) | 41.0 | 3.9 | 10.8 | 27.0 | 17.3 |

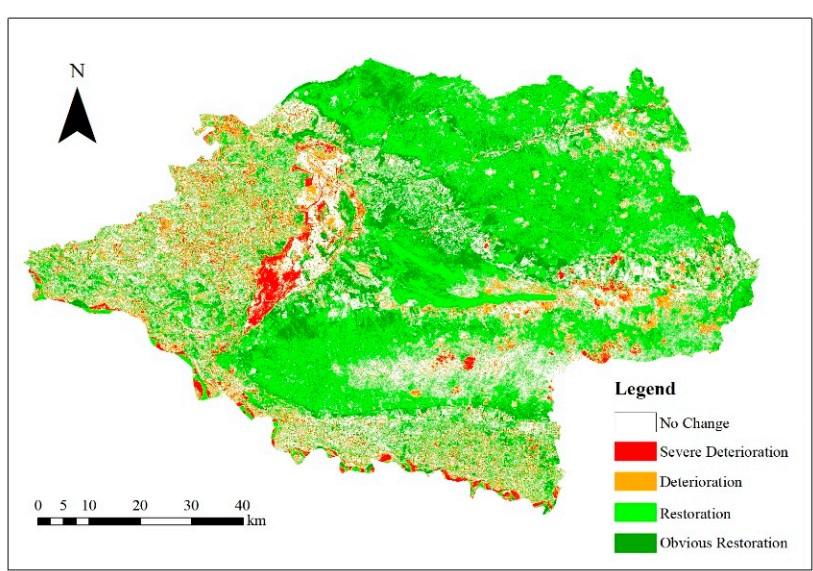

**Figure 5.** Changes in spatio-temporal patterns of desertification in the study area.

The area transformed into each type of desertification land in the Urat front flag is 66.9% of the total area of the Urat front flag. Additionally, the amount of transformation into each type of desertification land from high to low is: extremely serious desertification > moderate desertification > weak desertification > serious desertification > non-desertification. The transformation of desertification types in the Urat front flag mostly ranges from very serious to moderate, from serious to non-desertification, and from weak to non-desertification, covering 22.5045 km$^2$, 44.0478 km$^2$, and 319.2160 km$^2$, respectively. Nearly half of the serious desertification land is transformed into other types of land, and the area of other types of land transformed into serious desertification land is 540.7380 km$^2$, while the area of land converted from other types of land to

non-desertification is 309.3220 km$^2$, accounting for 18.2% of non-desertification land in 2020. This indicates that, as a result of the combined effect of the natural environment and human activities, the conversion of the Urat front flag to diverse forms of desertification has occurred on a larger scale and in a good direction. This underlines the fact that the Urat front flag has achieved remarkable results in desertification control under the auspices of the sand control project.

To gain a better understanding of the spatial dynamics of desertification in the study area from 2010 to 2020, changes in desertification spatial patterns were classified into five categories (Figure 5 and Table 4): severe deterioration (desertification increased by two or more levels), deterioration (desertification increased to the adjacent level), no change (desertification remained unchanged), restoration (desertification decreased to the adjacent level), and obvious restoration (desertification decreased by two or more levels).

Spatial statistics indicate that between 2010 and 2020, the distribution of areas with varying degrees of desertification in the Urat front flag remained stable. No change areas were sparsely distributed, primarily in the west (e.g., Uliang Suhai Fishery in Bayannur City and Xin'an Farm in Bayannur City) and the south (e.g., Baiyanhua Town). Deterioration and severe deterioration areas were mainly concentrated in the central west (e.g., Xishanzui Farm in Bayannur City and Xin'an Farm in Bayannur City) and south. Restoration and obvious restoration areas were mostly situated in the northern (e.g., Shetai Ranch, Sudulun Town, and Bayannur City) and central (e.g., Erdenbrag Sumu and Ming'an Town) areas. In addition, between 2010 and 2020, the degree of desertification remained stable in certain areas of the Urat front flag and did not change (10.8% of the study area), while the desertification aggravation and the recovery areas accounted for 41% and 44.3% of the total area, respectively. Among them, the desertification restoration area is 199.11 km$^2$ larger than the desertification aggravation area.

Meanwhile, according to the desertification land transfer matrix reported in Table 3, the desertification recovery area in the study area between 2010 and 2020 is primarily derived from the conversion of extremely serious to moderate, serious to non-desertification, and moderate desertification to non-desertification, whereas the desertification aggravation area is primarily derived from the conversion of serious desertification to moderate desertification.

### 3.7. Spatial Pattern Variation of Desertification Land

The dynamic changes in the spatio-temporal distribution of different types of desertification land between 2010 and 2020 were quantified using the standard deviation ellipse method, and the center of gravity migration model calculation. Equations (10)–(12) were used to determine the migration rates of different levels of desertification land centers of gravity in the Urat front flag from 2010 to 2020, respectively (Table 5). Additionally, the longitudinal and latitudinal coordinates of each center of gravity were plotted according to the spatial migration map of each center of gravity of desertification land between 2010 and 2020 (Figure 6).

**Table 5.** Change in the center of gravity for desertification land in the Urat front banner from 2010 to 2020.

| The Type of Desertification Land | Barycentric Coordinates for 2010 | | Barycentric Coordinates for 2020 | | Migration Distance during 2010–2020 | Rate of Migration during 2010–2020 |
|---|---|---|---|---|---|---|
| | X (°′″) | Y (°′″) | X (°′″) | Y (°′″) | D (km) | V (m/a) |
| Extremely serious | 235°39′14″ | 143°18′11″ | 20°11′44″ | 233°17′20″ | 27.8533 | 12.8942 |
| Serious | 315°17′17″ | 343°32′08″ | 131°25′08″ | 26°11′51″ | 4.9716 | 31.8593 |
| Moderate | 12°43′31″ | 114°38′03″ | 233°07′18″ | 206°45′01″ | 5.1422 | 13.8593 |
| Weak | 87°51′36″ | 106°55′13″ | 289°12′51″ | 319°18′30″ | 7.9386 | 21.3242 |
| Non-desertification | 299°46′48″ | 161°37′11″ | 73°14′44″ | 253°32′22″ | 2.0950 | 16.5944 |

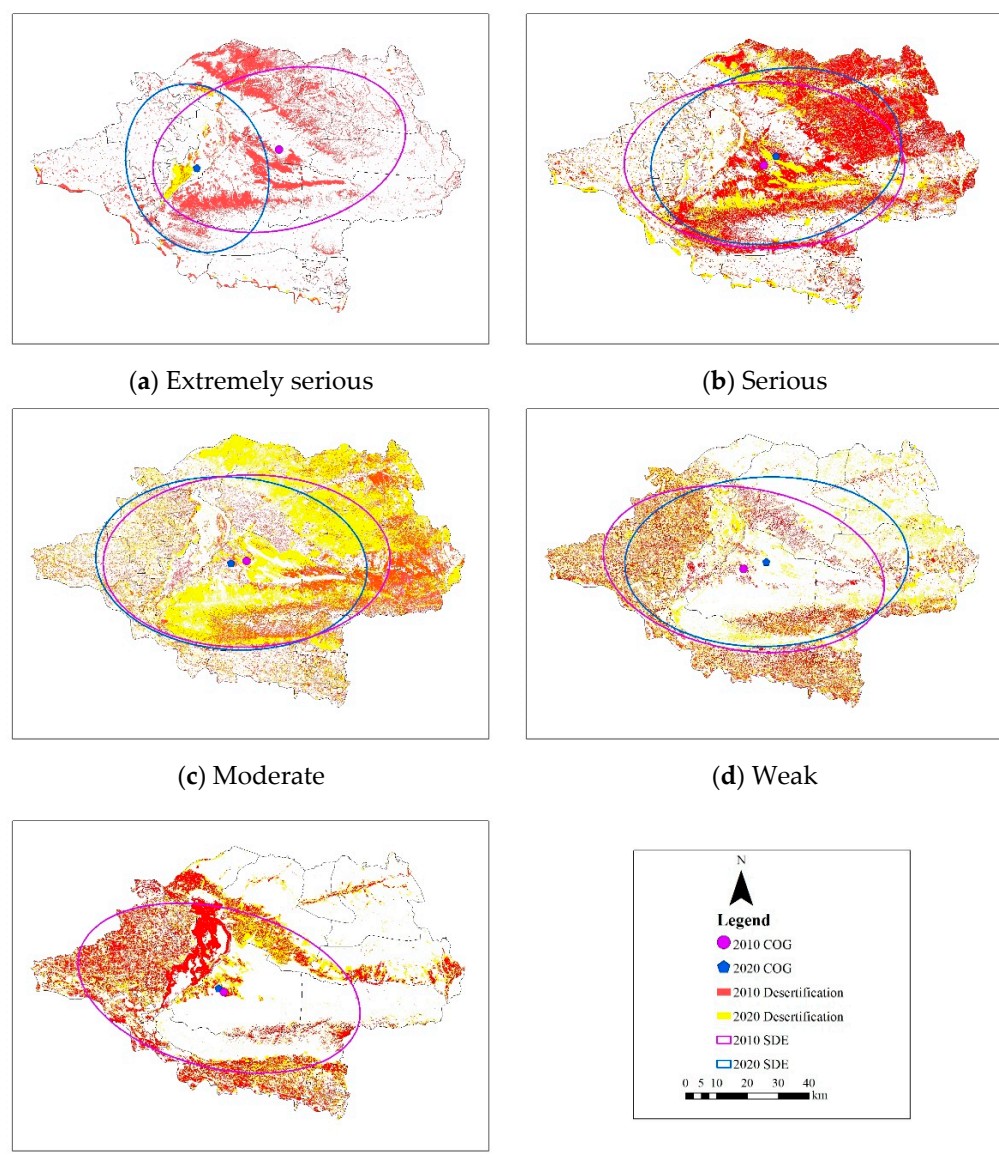

(**a**) Extremely serious

(**b**) Serious

(**c**) Moderate

(**d**) Weak

(**e**) Non-desertification

**Figure 6.** Center of gravity migration of desertification land in the Urat front banner from 2010 to 2020. Note: COG: center of gravity; SDE: standard deviation ellipse.

During the period of 2010–2020, the center of gravity of each desertification land has been migrated and altered dramatically. The extremely serious desertification land migrated 27.8533 km from the central southwest direction at a migration rate of 12.8942 m/a, indicating that the repeated and excessive livestock feeding and trampling in the Xin'an farm of Bayannur City and the Wuliang Suhai fishery of Bayannur City destroyed the community structure of forage grasses, reduced their quality, significantly decreased the edible forage grass yield, thinned and shortened the plants, and gradually decreased the coverage, resulting in concentrated patches of bare grass. Serious desertification land migrates 4.9716 km from the central part to the northeast, with a migration rate of 31.8593 m/a, mostly due to the low precipitation in the northeast region, drought, and water shortage. Moderate desertification land migrates 5.1422 km along the central part to the west at a migration rate of 13.8593 m/a, indicating that desertification has developed to the south and northeast of the oasis, at a migration rate of 13.8593 m/a. The main reason is that the soil structure and surface vegetation are destroyed by excessive or indiscriminate reclamation. Non-desertification land has the shortest distance of migration between 2010 and 2020, with

only 2.0950 km from the southeast to the northwest, at a migration rate of 16.5944 m/a. Deforestation of land is a result of anthropogenic activities such as deforestation and extensive groundwater extraction.

The spatial migration pattern of the center of gravity for different desertification types in the Urat front flag from 2010 to 2020 is depicted in Figure 6. Extremely serious desertification occurs primarily in the central and northern parts of the study area. Weak and non-desertification land is mostly distributed in the southwestern and northwestern parts of the study area, where it is mixed and intermingled. Serious and moderate desertification are mainly concentrated in the central, northeastern, and southeastern parts of the study area. The reason is that the geographical location and climatic environment of the study area lead to drought, low rainfall, and water shortages. The degree of desertification in the region is dominated by serious, weak, and non-desertification land types. As the spatial distribution patterns of desertification center of gravity and actual desertification land are essentially the same, changes in the centers of gravity of different types of desertification land can accurately reflect the spatial changes in desertification land.

## 4. Discussion

In order to further investigate the drivers of desertification land change from 2010 to 2020, the NPP values in 2010 and 2020 in Urat front flag were estimated using MODIS image data and CASA model and their spatial distribution was analyzed using ArcGIS.

NPP has the most intuitive way to convey the degradation of vegetation productivity and can be distinguished from the other forms of desertification land types. As illustrated in Figure 7a, the annual NPP per unit area in 2010 in the Urat front flag can reach a maximum of 306.3 gC/m$^2$, with an annual average value of 122.2 gC/m$^2$, in which the NPP of the Dashetai Ranch in Bayannur City, Zhongtan Farm in Bayannur City, the southwestern part of Dashetai Town, and some parts of Mingan Town is significantly higher than the NPP of vegetation in the western and central and northeastern regions. This phenomenon indicates that in the central and northern regions of the study area, the climate is arid, precipitation is scarce, vegetation is sparse, desertification is severe, and NPP is low. As illustrated in Figure 7b, the variation in NPP in 2020 is relatively obvious. The annual NPP can reach a maximum of 354.5 gC/m$^2$, while the annual average value is 159.5 gC/m$^2$, which exhibits an increase of 48.2 gC/m$^2$ compared with 2010, and the areas with strong vegetation growth were mainly concentrated in the southern, north-central, and east-central regions. In the eastern and southern parts of the study area, more farms, pastures, grasslands, forests, and water resources are reorganized, and the vegetation NPP is higher. There are obvious differences in spatial distribution over the past 10 years, and the results of desertification control are remarkable.

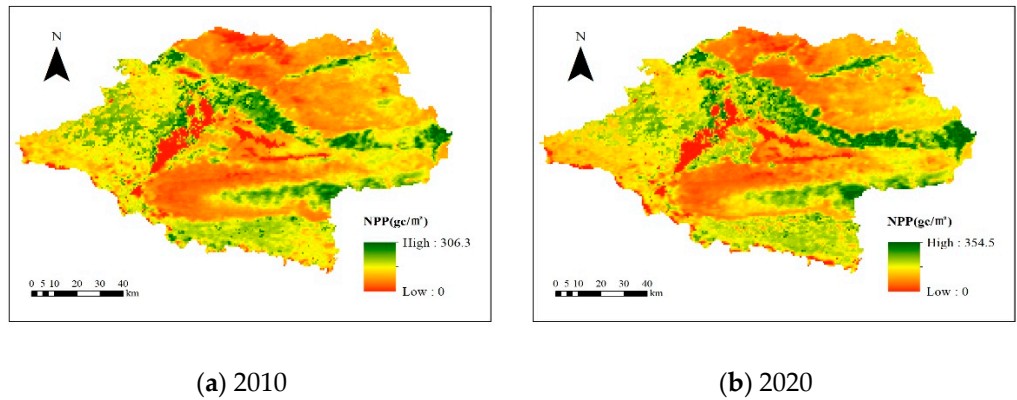

(**a**) 2010　　　　　　　　　　　　　　(**b**) 2020

**Figure 7.** Spatial distribution of vegetation NPP in Urat front flag.

According to earlier studies, changes in vegetation NPP result from a synergistic impact of human activities and climate change [62–67]. The annual average NPP of vegetation in the Urat front flag from 2010 to 2020 has a distinct spatial differentiation pattern (Figure 7),

with an overall growing distribution in a band from west to east, and the differences are related to hydrothermal conditions. The northwestern Bayannur City Xin'an Farm and the north-central Dashetai Town have a higher vegetation NPP due to grassland and woodland planting. The northeastern Xiaoshetai Town and parts of northern Sudulun Town have poor natural conditions and a relatively low vegetation NPP, and the south-central Erdenbrag Sumu is dominated by a stable continental climate with insufficient precipitation, sparse vegetation cover, and low vegetation NPP. The NPP of vegetation is also higher because the southern part is bordered by the Yellow River, with good climatic conditions and sufficient water resources. In summary, climate change is the primary factor influencing vegetation recovery in Urat former banner, while precipitation is the primary climatic factor controlling vegetation growth. Wet climatic conditions and sufficient precipitation are conducive to vegetation growth, and this effect is especially significant in arid and semi-arid areas. The effect of precipitation on vegetation is primarily through affecting vegetation photosynthetic efficiency and plant activity, thereby influencing vegetation organic matter production.

## 5. Conclusions

This paper analyzed the spatial and temporal evolution of land desertification in the Urat front flag during the last decade from the perspective of time and space, utilizing the Albedo–NDVI feature space method. It intends to conduct a scientific evaluation of the effect of ecological restoration projects and ecological security status implemented in the Urat front flag by investigating the spatial and temporal evolution laws of desertification, as well as to serve as a technical reference for national desertification control research. The study discovered that:

(1) The overall desertification status in the Urat front flag has improved, and the desertification land area for each grade has been altered to different degrees.

(2) Between 2010 and 2020, the desertification recovery area in the study area is primarily driven by the conversion of extremely serious to moderate, serious to non-desertification, and moderate desertification to non-desertification, whereas the desertification aggravation area is primarily driven by the conversion of serious desertification to moderate desertification.

(3) In descending order, the conversion rate of each type of desertification land area is as follows: extremely serious desertification > moderate desertification > weak desertification > serious desertification > non-desertification.

(4) The study area is arid with little rainfall and a water scarcity as a consequence of its geographic location and climatic environment, and the degree of desertification in the region is dominated by serious, weak, and non-desertification land types.

(5) The dynamic change of vegetation NPP is the consequence of the combined effects of climate change and human activities. Annual NPP per unit area in Urat front flag reached 306.3 gC/m$^2$ in 2010 and 354.5 gC/m$^2$ in 2020, a rise of 48.2 gC/m$^2$ each year since 2010, and the overall distribution is growing in a band from west to east.

**Author Contributions:** Conceptualization, Y.F., S.W., M.Z. and L.Z.; methodology, Y.F. and S.W.; Investigation, Y.F., S.W., M.Z. and L.Z.; data curation, L.Z.; formal analysis, Y.F., S.W. and M.Z.; writing—original draft preparation, Y.F.; writing—review and editing, S.W., M.Z. and L.Z.; resources, Y.F., S.W., M.Z. and L.Z.; project administration, Y.F. and S.W. All authors have read and agreed to the published version of the manuscript.

**Funding:** This research was funded by the National Natural Science Foundation of China (Project No.: 31700369).

**Institutional Review Board Statement:** Not applicable.

**Informed Consent Statement:** Not applicable.

**Data Availability Statement:** Data is contained within the article.

**Conflicts of Interest:** The authors declare no conflict of interest.

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
