# Peer review of "Monitoring of Land Desertification Changes in Urat Front Banner from 2010 to 2020 Based on Remote Sensing Data"

_water, doi:10.3390/w14111777_

Round 1
Reviewer 1 Report
Interesting topic and well structured, however, the following comments "few only are major" should be considered to improve the work:
Abstract:
- There is no need to mention the headlines in the abstract, please remove them (background, methods …etc.).
Introduction:
- Lines# 106-109: the objective is very short described, please provide more details about the aim of the study and highlight the key objectives.
Materials and methods
- Lines# 125 and 133: the links should be moved to the references please and replace them in the paragraph with reference numbers.
- In section: 2. Data source and its pre-processing: the author didn’t give a justification why using these algorithms, approaches or software. Please add coupe sentences to illustrate the characteristics of using such approaches and why they believe it is better than other methods or algorithms.
- Line# 146: The author mentioned using the MOD17A3 data obtained for the two years of 2010 and 2020” .. as far as the author/s claimed Ten years of monitoring and investigation 2010-2020, why have two years have been mentioned here for the MOD17A3 data? Please clarify this to avoid confusion.
- Figure 2. Albedo-NDVI feature space: what makes this general sketch important to be mentioned, please?
Results and analysis
- Lines# 290-291: “To ensure the accuracy of the results, 100 validation points were distributed uni-290 formly and randomly throughout the experimental area” … this sounds more related to the methods not to the results!
- Figure 4. Better resolution is required, the legend is unclear.
- Lines# 332-333: what is the reason for the significant reduction in desertification between 2020-2010? This needs to be mentioned, please.
- Table 3. please better organize lines 1 and 2 (years and the Extremely serious).
- Figure 6. Legends are unclear.
Reviewer 2 Report
The article is interesting for the readers of water (mdpi journal). However, some improvements a necessary in order to be accepted.
In the abstract please add some numerical results. This is important so as a readers have a clear view for the results just reading the abstract.
Once the specific topic is determined, the literature review needs to be performed further in-depth. Author’s needs to develop a clear logic on why the topic of this article is necessary based on the findings of the previous works that addressed the similar issue. Now, I don't find such a clear logical flow in the introduction.
In the last paragraph of the intro, please state the research gap answered by this research, also state the novelty points of the proposed approach according the similar results (novel points.
You results may discuss further comparing with similar researches regarding climate change and the importance of desertification for the environmental importanct areas eg
- Stefanidis, S., Alexandridis, V., & Ghosal, K. (2022). Assessment of Water-Induced Soil Erosion as a Threat to Natura 2000 Protected Areas in Crete Island, Greece. Sustainability, 14(5), 2738.
The reference must be corrected according the journal style guides.
Round 2
Reviewer 1 Report
The author/s have been well answered about most of the questions that have been addressed. and the piece of work have been developed. however one more tiny comment on the below conversation:
Point 7: Lines# 290-291: “To ensure the accuracy of the results, 100 validation points were distributed uni-290 formly and randomly throughout the experimental area” … this sounds more related to the methods not to the results!
Response 7: This is a good question. In the results and analysis, it is stated that "to verify the accuracy of the results obtained, 100 validation points were evenly and randomly distributed throughout the experimental area" because the theoretical knowledge of the experimental method is described in the second part, and the data processing process and operation are presented in the results and analysis, which will make the paper more logical and organized.
I am still believe this couple lines should be moved to the methods. hence I would leave it to the Editor to make the final judgment since the author/s opinion is different than mine.
Author Response
Thanks for your comments. I have revised the paper according to your comments.
Reviewer 2 Report
The article can be accepted in the current form
